# A network-centric approach to drugging TNF-induced NF-κB signaling

Nicolas A. Pabon[1], Qiuhong Zhang[1], J. Agustin Cruz[1], David L. Schipper[1], Carlos J. Camacho[1] & Robin E.C. Lee [1]

Target-centric drug development strategies prioritize single-target potency in vitro and do not account for connectivity and multi-target effects within a signal transduction network. Here, we present a systems biology approach that combines transcriptomic and structural analyses with live-cell imaging to predict small molecule inhibitors of TNF-induced NF-κB signaling and elucidate the network response. We identify two first-in-class small molecules that inhibit the NF-κB signaling pathway by preventing the maturation of a rate-limiting multiprotein complex necessary for IKK activation. Our findings suggest that a network-centric drug discovery approach is a promising strategy to evaluate the impact of pharmacologic intervention in signaling.

[1] Department of Computational and Systems Biology, School of Medicine, University of Pittsburgh, Pittsburgh, PA 15213, USA. These authors contributed equally: Nicolas A. Pabon, Qiuhong Zhang. These authors jointly supervised this work: Carlos J. Camacho, Robin E. C. Lee. Correspondence and requests for materials should be addressed to C.J.C. (email: CCamacho@pitt.edu) or to R.E.C.L. (email: RobinLee@pitt.edu)

A dynamic and complex network of interacting proteins regulates cellular behavior. Traditional target-centric drug development strategies prioritize single-target potency in vitro to modulate key signaling pathway components within the network and produce a desired phenotype. Target-centric strategies use biochemical assays to optimize specificity and affinity of small molecules for a protein class, such as protein kinases, or a specific enzyme. In some cases, an effective inhibitor is comparable with gene knockdown (KD) that reduces or completely removes the target protein from the network. However, given that pleiotropy is prevalent among disease-associated proteins, compounds that disrupt specific protein–protein interactions (PPI) while leaving others intact are attractive, especially when complete disruption is detrimental to the cell[1,2]. Small molecules are a promising class of PPI inhibitors to perturb signaling networks in vivo, but they are technically difficult to identify and assess. Instead, many PPI inhibitors are derived from competitive peptides with challenging cell permeability and pharmacokinetic properties[3].

Tumor necrosis factor (TNF)-induced nuclear factor (NF)-κB signaling is an example of a tightly regulated and therapeutically relevant pathway that has resisted target-centric drug discovery. TNF is an inflammatory cytokine that initiates dynamic intracellular signals when bound to its cognate TNF receptor (TNFR1). In response to TNF, the IκB-kinase (IKK) complex is rapidly recruited from the cytoplasm to polyubiquitin scaffolds near the ligated receptor where it is activated through induced proximity with its regulatory kinase, TAK1[4–10]. When fully assembled, the mature TNFR1 complex (Fig. 1a) is a master regulator of inflammation-dependent NF-κB signaling. NF-κB inhibitor proteins (IκB) are degraded soon after phosphorylation by activated IKKs, and the NF-κB transcription factor accumulates in the nucleus to regulate TNF-induced transcription. Since changes in the subcellular localization of IKK and NF-κB transmit stimulus-specific information[11–14], these dynamic features can be used to demonstrate pharmacologic alterations to inflammatory signaling[15].

Chemicals that modulate inflammation-dependent IKK and NF-κB signals are of considerable therapeutic interest. Activated NF-κB regulates the expression for hundreds of genes that mediate signals for inflammation, proliferation, and survival[16–21] and its deregulation is linked to chronic inflammation in addition to the development and progression of various cancers[22–25]. As pleiotropic proteins, IKK and NF-κB are poor targets for inhibitors because they provide basal activity as survival factors independent of inflammatory signaling[26] and their genetic disruption can be lethal[27,28]. The complexity of the pathway and the difficulty of modulating specific PPIs in vivo exacerbates the challenges of drugging this pathway in the cell[29]. Not surprisingly, there are no clinically approved small-molecule inhibitors of NF-κB pathway components.

An alternative network-centric strategy is to predict small molecules that act on rate-limiting PPIs in the signaling pathway in silico and screen them for phenotypes associated with pathway disruption in vivo. Although complete disruption of IKK and NF-κB can have damaging effects on the cell, their dynamics in response to disease-associated inflammatory signals are influenced by >50 other proteins. Thus the broader NF-κB network contains numerous entry points for chemicals to impinge on the pathway. Here we use machine learning with gene expression (GE) data to provide a synoptic list of likely small-molecule inhibitors of the NF-κB pathway. For a well-defined molecular network, we show that pathway-specific inhibitors can be predicted from transcriptomic alterations that are shared between (i) exposure to small molecules and (ii) genetic KDs of the pathway components. Through molecular docking, we reduce the list of predicted compounds and suggest a mechanism of action, evaluating bioactivity using live-cell experiments that monitor signaling dynamics in single cells. We find two first-in-class small molecule inhibitors of the pathway that limit PPIs upstream of IKK recruitment and inhibit TNF-induced NF-κB activation. Our results combine to demonstrate a valuable network-centric systems biology approach to drug discovery.

## Results

**Identifying candidate inhibitors of NF-κB signaling.** To demonstrate a network-centric strategy for targeting TNF-induced NF-κB signaling, we focused on differential GE signatures from the NIH Library of Integrated Network-Based Cellular Signatures (LINCS) L1000 dataset[30]. We compared transcriptional profiles between genetic KDs of proteins in the NF-κB signaling pathway and responses of the same cell types to thousands of distinct bioactive compounds. Using a random forest classification model trained using Food and Drug Administration (FDA)-approved drugs, we identified compounds whose transcriptomic perturbations resembled genetic disruption. For each compound, the probability of a compound–protein interaction was evaluated in terms of several attributes, including direct correlation with the KD signatures and indirect correlations with KD signatures of other proteins in the network for ≥4 cell lines (see ref. [31] for detailed explanation). In the context of a protein interaction network, disruption of a physical target by a drug can cause similar GE profiles as inhibition of downstream or upstream genes in the same subnetwork. Hence, a compound that disrupts TRADD or TRAF2 in Fig. 1a might have similar signatures to the KD of genes in the pathway such as TNFR1, UBC, or NEMO (see below). Here we leverage this guilt by association, which suggests that chemical inhibition acts broadly within a signaling subnetwork (Supplementary Fig. 1), to drug the NF-κB signaling pathway.

A PPI inhibitory peptide that competes with recruitment of catalytic IKK subunits at ubiquitin scaffolds was previously shown to inhibit inflammatory NF-κB activation and disease progression in a murine model for inflammatory bowel disease[26,32]. We reasoned that any compounds that disrupt the mature TNFR1 complex, particularly at the level of TRADD, TRAF2, and RIP1, will prevent TNF-mediated IKK recruitment and nuclear translocation of NF-κB. Transcriptional signatures for 717 unique compounds showed strong correlations with genetic KDs of TRADD, TRAF2, and RIP1. From this initial set, we identified potential pathway inhibitors as compounds that also correlated with genes in the mature TNFR1 complex (Fig. 1a). Specifically, we ranked candidate inhibitors by their mean Pearson correlation with NF-κB-pathway KDs to assist selection of compounds for additional screening (Supplementary Fig. 2).

**Targeting core PPIs in the mature TNFR1 complex.** Molecular docking was used to further refine the list of candidate compounds and predict mechanism of action against proteins in the TNFR1 signaling complex. The 717 candidate molecules described above were docked with domain structures available in the PDB for TRAF2, TRADD, and RIPK1. TRAF2 emerged as a promising target because, contrary to the other proteins, co-crystal structures of TRAF2 are available. Namely, the PPI between TRAF2 and both TRADD (PDB code 1F3V[33]) and a TNFR2 peptide (PDB code 1CA9[34]) have been characterized. Both co-crystals indicate a well-defined binding site, which was used to visually screen the top scoring compounds based on both Pearson correlation and binding scores ($n = 180$ compounds; see Supplementary Fig. 2). Three compounds whose binding modes replicate native contacts in the TRADD-TRAF2 protein complex

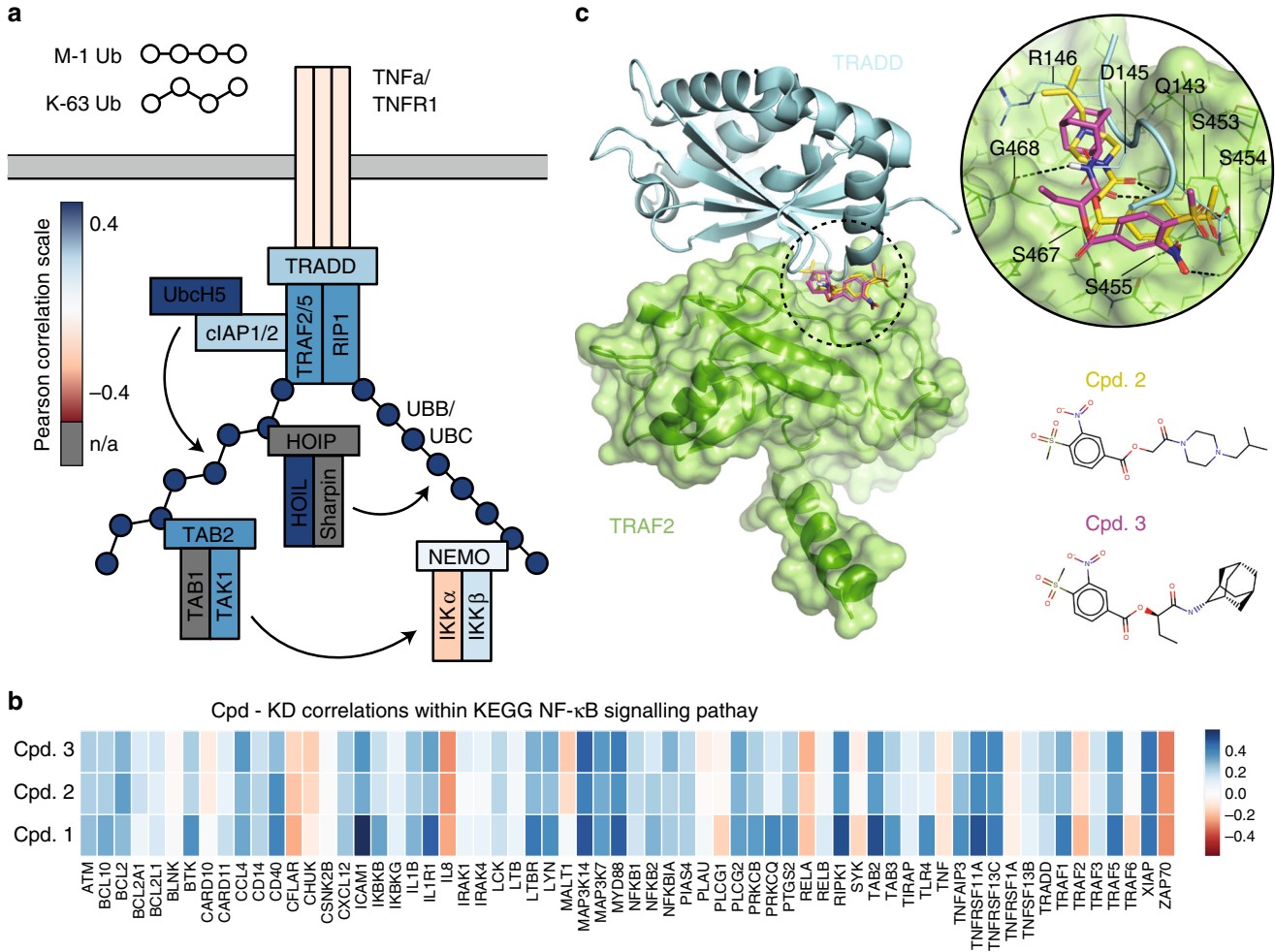

**Fig. 1** Transcriptional responses to compounds correlate with knockdowns of NF-κB pathway genes. **a** Schematic of the mature tumor necrosis factor (TNF) receptor 1 (TNFR1) complex, a cytoplasmic multi-protein complex that assembles following ligation of TNF to TNFR1. The color for each protein species in the complex is the average Pearson correlation between gene expression profiles for the species' genetic knockdown and the transcriptional response to compounds 2 and 3. **b** Correlation between transcriptomic perturbations by compounds 1, 2, and 3 and the knockdown of genes functionally involved in NF-κB according to the KEGG PATHWAY Database. Pearson correlation color scale is shown (right). **c** Unbiased molecular docking predicts binding of compounds 2 (yellow) and 3 (magenta) to the TRADD-binding interface of TRAF2. Hydrogen bonds with key TRAF2 interface residues are indicated by dotted lines. Source data are provided as a Source Data file

were selected for testing: (1) BRD-K43131268, (2) BRD-K95352812, and (3) BRD-A09719808. For compounds 1, 2, and 3 respectively, predicted targets from our genetic KD GE dataset[31] included: TRAF2, UBC, NFKB1, and RIP1; TRAF6, NEMO, TRAF2, NFKB1, UBC, TAB2, and IKKβ; and NFKB1, TRAF2, UBC, UBB, and NEMO. Furthermore, compounds 2 and 3 showed significant correlations with both HOIL, TAK1, cIAP1/2, and UbcH5 KDs (Fig. 1a) and the corresponding transcriptional profiles for gene KDs in the NF-κB pathway (Fig. 1b). Compounds 2 and 3 also have similar chemical structures (Fig. 1c), strongly suggesting a similar mechanism of action.

Compounds 2 and 3 formed hydrogen bond contacts with TRAF2 residues S453, S454, S455, and S467, which are predicted to compete with TRADD interface residues Q143, D145, and R146 based on the co-crystal (Fig. 1c). Compound 3 is predicted to bind stronger due to the extra hydrogen bond formed by its amide group with TRAF2 residue G468. Of note, all these TRAF2 residues are conserved in TRAF5. Competitive binding should disrupt the native TRADD–TRAF2/5 PPI interface and could prevent maturation of the full TNFR1 signaling complex by promoting dissociation or allosteric stabilization of a non-native conformation. The predicted binding mode of compound 1 is less

specific and did not form any of the contacts described above (Supplementary Fig. 3).

To test whether the compounds interact with TRAF2 in vitro, we measured the thermal stability of purified TRAF2 in the presence of each compound. Thermal shift assays showed that compounds 2 and 3, respectively, exert a subtle-to-moderate dose-dependent stabilizing effect on full-length TRAF2 (Fig. 2a, b), suggesting direct compound–protein binding. In contrast, compound 1 did not show a clear trend (Supplementary Fig. 4). We note that the observed thermal shifts are consistent with the relatively small stabilizing effect that the compounds are expected to exert on the stable trimer formed by the soluble full-length TRAF2 protein[34]. Together, these data suggest that compounds 2 and 3 may impinge on TNF-induced signaling.

**Small molecules disrupt TNF-induced NF-κB dynamics.** We set out to determine whether the compounds are effective inhibitors of NF-κB signaling in living cells. For this, the endogenous gene locus for the transcriptionally active RelA subunit of NF-κB was modified using CRISPR/Cas9 to encode a fluorescent protein (FP) fusion in U2OS cells (Supplementary Fig. 5), a cell line that forms

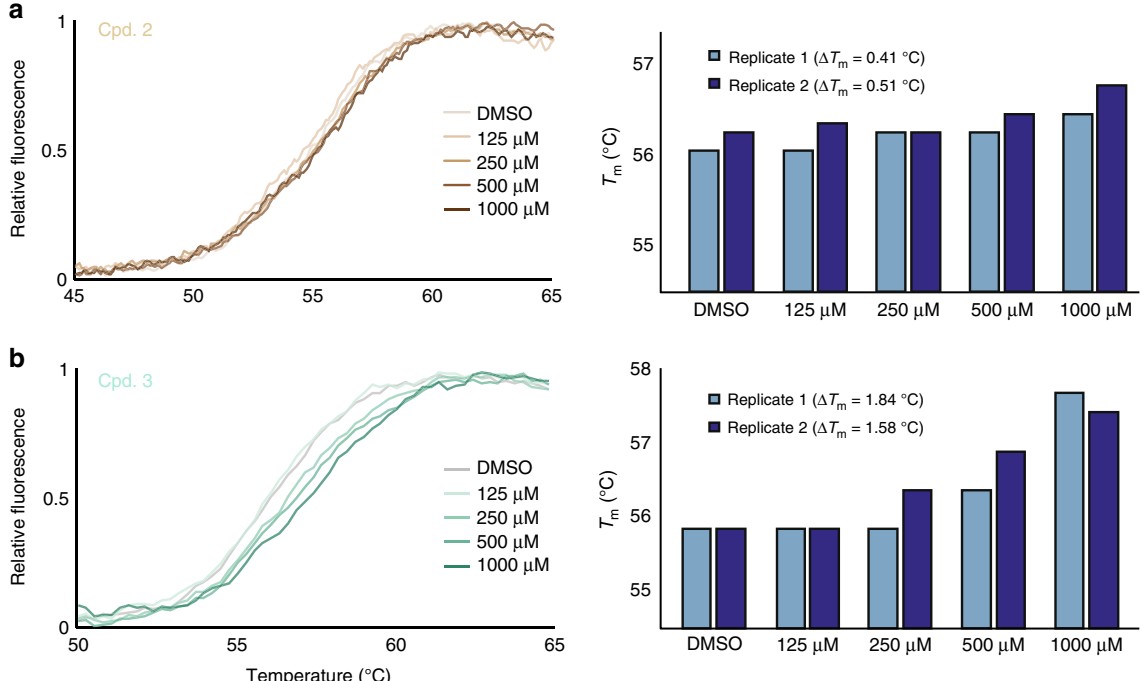

**Fig. 2** Thermal shifts indicate dose-dependent stabilization of TRAF2 by compounds 2 and 3. Normalized melt curves (left) and melting temperature ($\Delta T_m$; right) of full-length TRAF2 were recorded in the presence of dimethyl sulfoxide or indicated concentrations of **a** compound 2 (tan lines) and **b** compound 3 (teal lines). The rightward shift of the melt curve in the presence of compounds, quantified by the $\Delta T_m$ in two replicate experiments, suggest increased thermal stability of the protein–compound complex. Source data are provided as a Source Data file

IKK-recruiting polyubiquitin scaffolds in response to TNF[13]. Responses of single cells exposed to TNF showed transient and variable translocation of NF-κB into the nucleus when measured from time-lapse images (Fig. 3a, Supplementary Movie 1), comparable with other human cancer cell lines that express FP-RelA fusions[11,14,35]. When cells were pretreated with compounds 2 and 3 before exposure to TNF, nuclear mobilization of NF-κB was reduced with increasing concentration of the inhibitory compound (Fig. 3b).

To quantify the compounds' effect on NF-κB dynamics, each single-cell trajectory was decomposed into a series of descriptors (Fig. 3c) that transmit information within the cell about extracellular cytokine concentrations[14]. Descriptors of NF-κB dynamics that transmit the most information about TNF, including the area under the fold change curve (AUC) and the maximum fold change (Max)[14], were significantly less when cells were pretreated with 10 μM of the compound 2 or 3 before addition of TNF (Fig. 3d). Other descriptors showed a similar pattern of inhibition when exposed to 10 μM of either compound prior to TNF stimulation (Supplementary Figs. 6 and 7). By contrast, aside from subtle alterations to the rates of nuclear NF-κB mobilization, compound 1 did not significantly alter the overall TNF-induced dynamics of nuclear NF-κB (Supplementary Fig. 8). These data suggest that compounds 2 and 3 restrict the signaling network upstream of NF-κB activation with low micromolar potency (Supplementary Fig. 9).

Compounds 2 and 3 also showed significant correlations (Fig. 1b) with ubiquitination machinery and kinases, including IKK, that are common to basal cellular processes and inflammatory responses[36]. Interleukin-1 (IL-1) is one such inflammatory cytokine that activates NF-κB via the functional IKK complex but independent of interactions between TRADD and TRAF2. Instead, IL-1 utilizes TRAF6 that does not share any of the four serine residues (S453, S454, S455, and S467; Fig. 1a) identified as the binding substrate of our compounds. Consistent with this

observation and in contrast with the TNF response, IL-1-induced dynamics of nuclear NF-κB were indistinguishable between cells pretreated with compounds 2 or 3 and IL-1-only control cells (Fig. 4 and Supplementary Fig. 10). Furthermore, cytotoxicity analysis and assessment of IKKβ kinase activity in vitro demonstrated that compounds 1, 2, and 3 have low cytotoxicity and no direct inhibitory activity over IKKβ kinase activity at the concentrations used in this study (Supplementary Figs. 11 and 12). Together our results demonstrate that the IKK and NF-κB systems are intact in cells exposed to the compounds and suggest that the mode of action for both compounds is directed specifically at the level of the mature TNFR1 complex.

**Small molecules prevent formation of the mature TNFR1 complex.** Induced proximity between IKK and other regulatory factors within the mature TNFR1 complex is essential for TNF-induced NF-κB activation and may be perturbed in cells exposed to compounds 2 and 3. To test this hypothesis, and directly observe the penultimate recruitment of IKK to the TNFR1 complex, we used CRISPR/Cas9 to target the γ-subunit of IKK (also known as NEMO) for FP fusion and live-cell imaging in U2OS cells (Supplementary Fig. 13).

FP-IKK was diffuse within the cytoplasm of unstimulated cells and rapidly localized to punctate structures near the plasma membrane after exposure to TNF (Fig. 5a; Supplementary Movie 2). Because a key role of the TNFR1 complex is to recruit and activate IKK at ubiquitin scaffolds[10], detection of FP-IKK puncta can be used to measure maturation of the complex in living cells. The number of FP-IKK puncta in single cells peaked at 15 min and dissolved within an hour of TNF stimulation (Fig. 5b). Although the recruitment and dissolution dynamics of FP-IKK are prolonged when compared with a previous study that overexpressed a fusion of mouse IKKγ in U2OS cells[13], they are otherwise qualitatively similar.

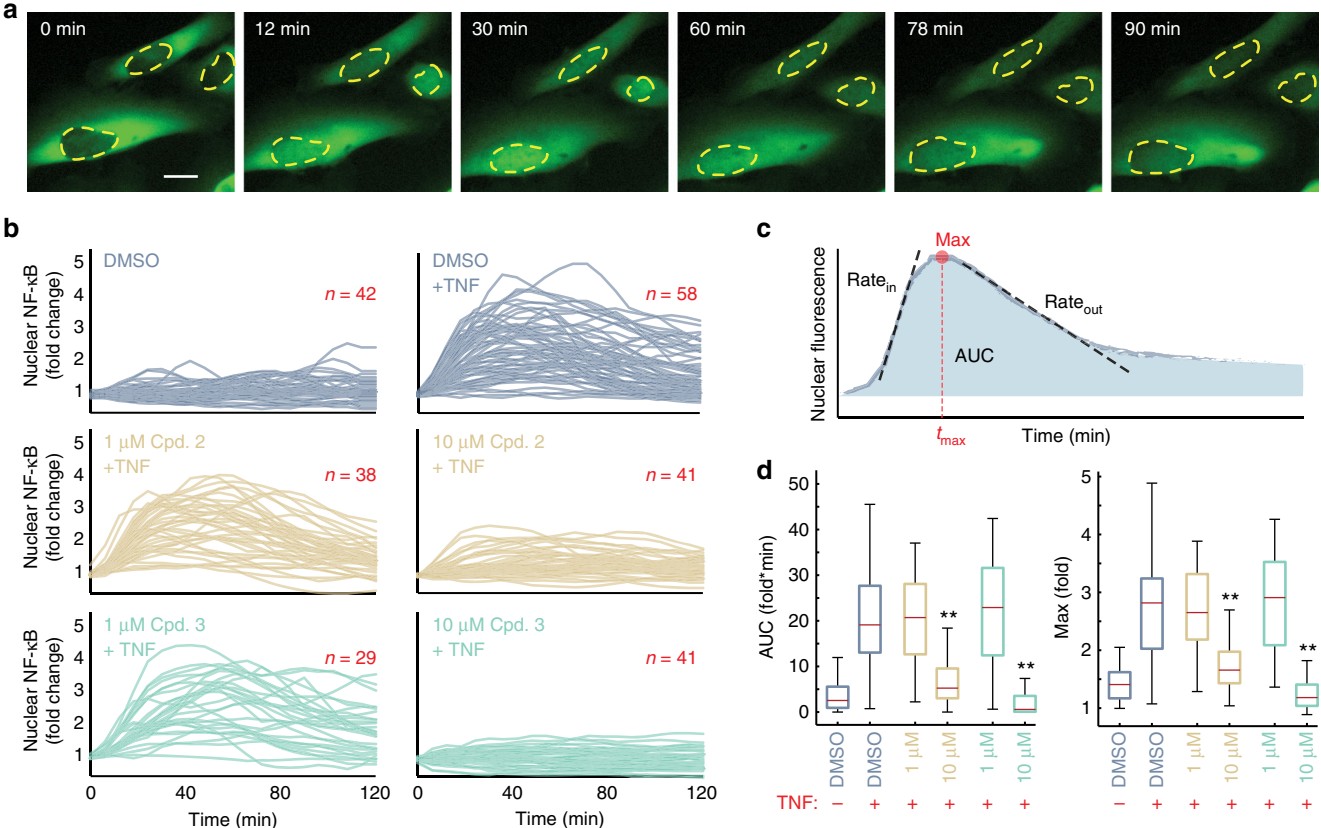

**Fig. 3** Predicted disruptors of NF-κB signaling reduce nuclear NF-κB translocation in response to TNF. **a** Time-lapse images of fluorescent protein (FP)-RelA expressed from its endogenous gene locus in U2OS cells exposed to TNF. The nuclear subcellular compartment is indicated with a broken yellow line. See also Supplementary Movie 1; Scale bar 20 μm for all. **b** Single-cell time courses of nuclear FP-RelA measure the change in the nuclear abundance of NF-κB in response to the indicated TNF conditions for cells pre-exposed to either dimethyl sulfoxide (blue lines), compound 2 (tan lines), or compound 3 (teal lines). Red numbers indicate the number of single-cell trajectories in each condition. **c** Descriptors used to quantify single-cell responses. AUC, Max, and $t_{max}$, respectively, describe the area under the curve, the maximum, and the time of maximal nuclear FP-RelA fluorescence. Rate$_{in}$ and Rate$_{out}$ describe the maximal rate of nuclear entry and exit. **d** Box (first and third quartile) and whisker (1.5 times interquartile range) plots showing the condition-specific variation for descriptors of nuclear FP-RelA localization. Indicated conditions and the number of single-cell time courses measured from **b** for each descriptor. Red center line indicate the median; red minus and plus symbols, respectively, indicate the absence or presence of TNF; double stars indicate $p$ value $<10^{-6}$ (two-tailed), based on permutation test (Supplementary Fig. 7). Source data are provided as a Source Data file

Consistent with our observations for NF-κB, the number of TNF-induced puncta were greatly reduced in cells that were pretreated with compounds 2 or 3 before exposure to TNF (Fig. 5b). Unexpectedly, the compounds also reduced the overall expression level of IKKγ (Supplementary Fig. 14) through an unknown mechanism that may relate to TRAF-dependent ubiquitination cascades that regulate the ambient stability of other NF-κB-inducing kinases[37]. Overall, the absence of IKKγ mobilization in TNF-stimulated cells indicate that micromolar concentrations of compounds 2 and 3 prevent a key proximity-induced mechanism provided otherwise through assembly of the mature TNFR1 complex.

## Discussion
Taken together, our results show that compounds 2 and 3 inhibit the TNF-induced NF-κB signaling pathway by limiting the formation of the mature TNFR1 complex. The mode of action of the compounds is specific to the TNF response, leaving intact core molecular components of the NF-κB pathway that are co-opted by other biological processes including responses to other inflammatory stimuli. We also highlight the broader effects of

disrupting a pathway component within the larger network, including the downregulation of IKKγ protein expression, and the limitations of single-target molecular modeling as a basis for drug design. The regulatory complexity of the NF-κB signaling pathway, which enables highly specific and stimulus-dependent transcriptional responses, also confounds drug discovery efforts that do not account for network-scale responses to chemical disruption. Consequently, successful therapeutic intervention in complex signaling pathways may require a network-centric strategy guided explicitly by a compound's anticipated effects on signaling dynamics as a pharmacologic target[15].

Correlations in GE signatures and single-cell experiments can be used to respectively predict and validate the network effects of bioactive compounds, and structural analysis can further inform on their mechanism of action. Here our models suggest that compounds 2 and 3 destabilize interactions between TRADD and TRAF family proteins. Mechanistically, disruption at this upstream junction will preclude ubiquitin scaffold assembly and rationalizes our data. In addition to the live-cell data, these include the correlations observed between the compounds and KDs of UBB, UBC, and other signaling proteins that are recruited to these polyubiquitin chains (see Fig. 1a), such as IKK and other

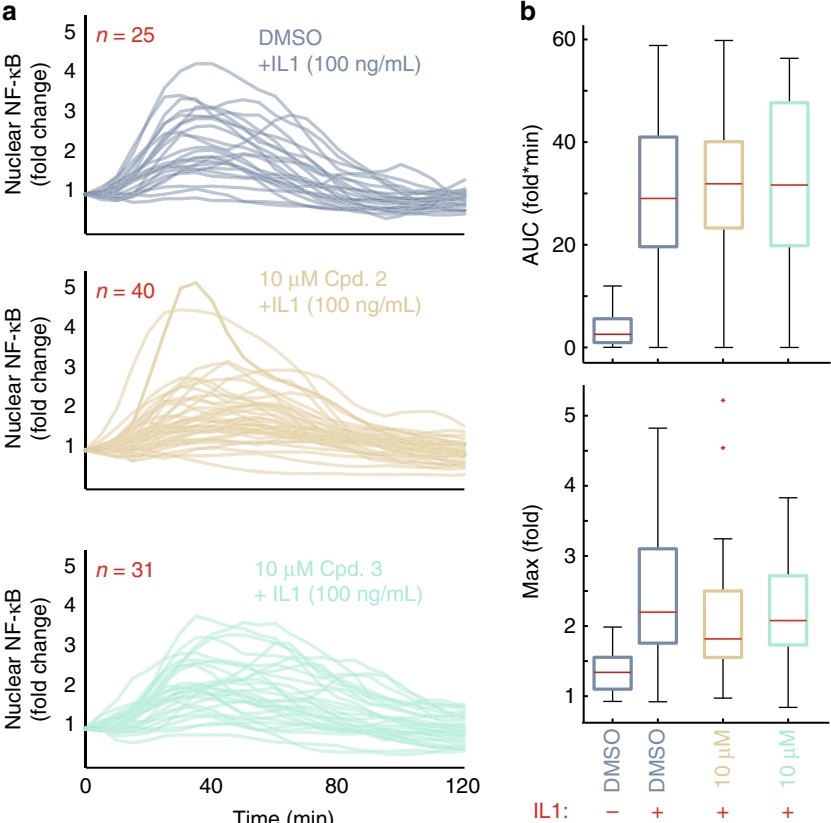

**Fig. 4** Compounds do not alter dynamics of nuclear NF-κB in response to IL-1. **a** Single-cell time courses of nuclear fluorescent protein (FP)-RelA quantified in cells exposed to 100 ng/mL IL-1 in addition to either dimethyl sulfoxide (top, blue lines), 10 μM compound 2 (middle, tan lines), or 10 μM compound 3 (bottom, teal lines). Red numbers indicate the number of single-cell trajectories for each condition. **b** Box (first and third quartile) and whisker (1.5 times interquartile range) plots showing descriptors for the area under the fold change curve (top) and the maximum fold change of nuclear FP-RelA (bottom) for the indicated conditions and number of single-cell time courses measured from **a** do not show significant changes based on permutation tests (Supplementary Fig. 10). Red center line indicates the median, outliers marked with red dots; red minus and plus symbols, respectively, indicate the absence or presence of IL-1. Source data are provided as a Source Data file

upstream regulators. A limitation is that identified compounds may also have alternative effects on other signal transduction pathways that are not explicitly considered here. Overall, by identifying two independent compounds that converge on similar genomic and functional phenotypes strongly suggests that the NF-κB pathway is specifically disrupted.

Although the LINCS dataset does not explicitly report the transcriptional response of cells to TNF in the presence of chemical or genetic perturbations, compounds that impinge on TNF-induced dynamics could still be inferred using a machine learning algorithm with prior knowledge of the signaling network. This pipeline therefore represents an alternative strategy to single-target-based drug discovery that can be more generally applied to discover novel inhibitors of protein subnetworks in a variety of signaling pathways. Because mechanism of action is not constrained a priori, it is possible to discover a chemical agent that disrupts multiple points in the same protein subnetwork or to predict chemical combinations that produce specific network-level responses. It is unlikely that the magic bullet drug discovery paradigm will uncover the full therapeutic potential of compounds that modulate PPIs and dynamic intracellular signals, such as the TNF-induced NF-κB signaling pathway. Rather, more effective drug development efforts may require approaches like the one presented here that embrace the complexity of regulatory networks and dynamic phenotypes associated with their disruption.

## Methods

**Analysis of GE data**. Gene KD and compound treatment GE signatures were extracted from the NIH LINCS L1000 Phase I and Phase II datasets (Gene Expression Omnibus accession IDs: GSE70138 and GSE92742). We collected signatures for the 1680 small molecules and 3104 gene KD experiments that had been performed in at least 4 of the 7 most common LINCS cells lines (A549, MCF7, VCAP, HA1E, A375, HCC515, HT19). We hypothesized that compounds that disrupt the TNF-inducible NF-κB signaling pathway should produce similar network-level effects, and thus similar differential GE signatures, to genetic KDs of proteins in the pathway. Thus, for each compound–KD signature pair in our dataset, we computed several cell-specific quantitative features, most importantly: direct correlation is the Pearson correlation coefficient between the compound treatment and the gene KD expression signatures in the given cell line, and indirect correlation is the fraction of the KD protein's interaction partners, as defined by BioGrid[38], whose respective KD signatures were highly correlated with the compound signature. Three additional features, quantifying baseline drug activity in the cell and the maximum and average compound-induced differential expression levels of NF-κB pathway proteins[31], were also calculated and used in the subsequent classification.

Using a Random Forest (RF) classifier trained in the expression signatures of 152 FDA-approved drugs with known mechanism(s) of action[31], features for every compound–KD pair ($n = 5,214,720$) were used to predict the probability that the compound would inhibit the KD protein's interaction network. The top 100 predicted interactions for each compound were extracted, and compounds whose predicted targets were enriched in TNF-induced NF-κB signaling genes were collected for structural analysis.

**Structural analysis**. For structural docking of RF-predicted inhibitors, representative crystal structures of TNF-inducible NF-κB signaling proteins (Supplementary Fig. 1) were mined from the PDB[39], optimizing for sequence coverage, structural resolution, and structural diversity. Domain structures were available for

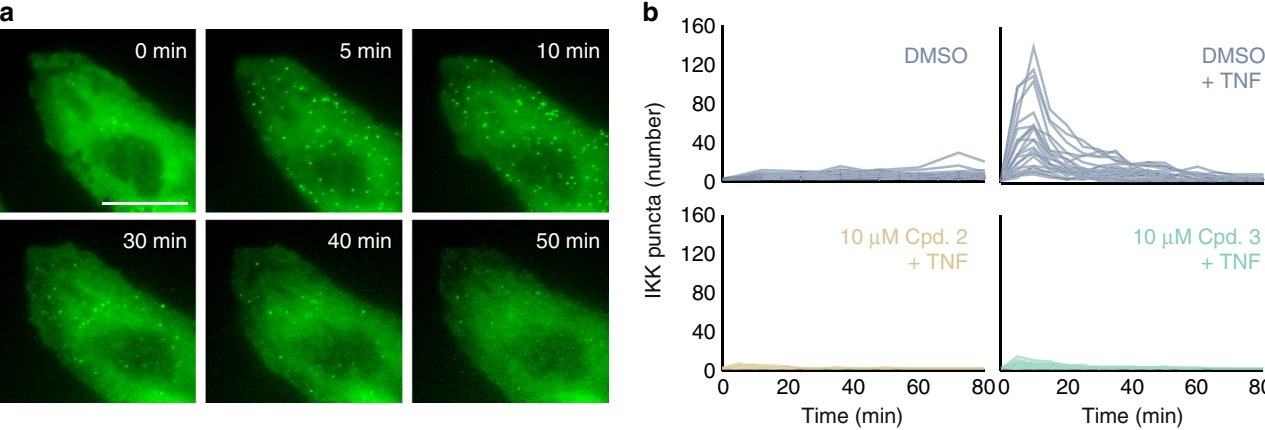

**Fig. 5** Compounds limit the formation of IKKγ puncta in TNF-stimulated cells. **a** Time-lapse images of fluorescent protein (FP)-IKK expressed from its endogenous gene locus in U2OS cells exposed to TNF. Scale bar is 20 μm for all. See also Supplementary Movie 2; **b** Single-cell time courses for the number of FP-IKK puncta in cells stimulated with the indicated TNF conditions for cells pre-exposed to either dimethyl sulfoxide (blue lines), compound 2 (tan lines), or compound 3 (teal lines). In all TNF conditions, a concentration of 100 ng/mL was used. Source data are provided as a Source Data file

all proteins in Fig. 1a with the exception of IKKα. Potential small-molecule binding sites on each protein structure were identified by clustering the output of computational solvent mapping software FTMap[40]. RF-predicted inhibitors were docked to predicted binding sites on each protein structure using smina[41] and a prospectively validated pipeline[42,43]. Generic versions of the three promising candidate inhibitors of TRAF2, which showed both biophysical complementarity and broad spectrum transcriptomic correlations with KDs in the pathway, were purchased from MolPort for experimental validation. Molport IDs were MolPort-000-763-757, MolPort-004-495-831, and MolPort-004-588-414 for compounds 1, 2, and 3 respectively. Notably, because of commercial availability, Molport versions of compounds 2 and 3 had minor modifications (see Supplementary Fig. 15) that do not alter their predicted binding profiles.

**Thermal shift assay and analysis**. TRAF2–compound interactions were measured by fluorescence-based thermal shift using an Applied Biosystems ABI QuantStudio(TM) 6 Flex System. All assay experiments used 1 μM GST-TRAF2 (Rockland) per well and 2× Sybro Orange (Invitrogen) in a buffer containing 50 mM HEPES, pH 7.5, 150 Mm NaCl in a total reaction volume of 15 μL in 384-well plates. Compounds were diluted with dimethyl sulfoxide (DMSO), and each reaction had a final DMSO concentration of 1.5%. PCR plates were covered with optical seal, shaken, and centrifuged after protein and compounds were added. The instrument was programmed in the Melt Curve mode and the Standard speed run. The reporter was selected as Rox and None for the quencher. Each melt curve was programmed as follows: 25 °C for 2 min, followed by a 0.05 °C increase per second from 25 °C to 99 °C, and finally 99 °C for 2 min. Fluorescence intensity was collected continuously. In the Melt Curve Filter section, X4 (580 ± 10)−M4 (623 ± 14) was selected for the Excitation Filter–Emission Filter. The raw data were extracted in MS-Excel format. Each melt curve was normalized between 0 and 1 and the midpoint of the curve was used to determine the melting temperature.

**Establishing EGFP-RELA/IKKγ CRISPR Knock-in cells**. The RelA repair template consisted of DNA sequences for a left homology arm (LHA −544 bp, chromosome 11_65663376–chromosome 11_65662383) followed by an enhanced green fluorescent protein (EGFP) coding sequence with a start codon but no stop codon and a sequence encoding 3x GGSG linker followed by a right homology arm (RHA +557 bp, chromosome 11_65662829–chromosome 11_65662276) were assembled from plasmids synthesized by GeneArt. Synonymous mutations that are not recognized guide RNAs were introduced to prevent interaction the repair template and Cas9. IKBKG DNA sequences for LHA (−861bp, chromosome X 154551142–chromosome X 154552002) and RHA (+797 bp, chromosome X_154552006–chromosome X 154552798) were amplified from Hela genomic DNA using the following primer pairs: IKBKG_LHA_F: 5′GGG CGA ATT GGG CCC GAC GTC GTT TCA CCG TGT TAG CCA GG3′, IKBKG_LHA_R: 5′CAC ATC CTT ACC CAG CAG CAG A3′; IKBKG_RHA_F: 5′AGA GTC TCC TCT GGG GAA GC3, IKBKG_RHA_R: 5′CCG CCA TGG CGG CCG GGA GCA TGC GAC GTC AGT CTA GGA AAG AAC TCC CCA GTC3′. To generate the fragment containing EGFP overlapping with LHA and RHA, we synthesized the sequence from GeneArt, then we amplified the sequence containing EGFP with the primer pairs: IKBKG_EGFP_F 5′TCT GCT GGG TAA GGA TGT G3′, IKBKG_EGFP_R 5′GCT CTT GAT TCT CCT CCA GGC AG3′. After PCR products were purified,

the fragments LHA, RHA, and EGFP were cloned to pMK plasmid that was digested with AatII by gibson assembly from NEB.

The guide RNAs were designed by the CRISPR Design Tool (http://crispr.mit.edu). Oligonucleotide pairs Rel A sg1 (top): 5′-CACCGCTCGTCTCTGTAGTGC ACGCCG-3′, Rel A sg1 (bottom): 5′-AAACCGGCGTGCACTACAGACGAGC-3′; RELA Sg2 (top) 5′-CACCGAGAGGCGGAAATGCGCCGCC-3′, RELA Sg2 (bottom) 5′-AAACCGCGGCGCATTTCCGCCTCTC-3′; IKBKG Sg1 (top) 5′-CA CCGCAGCAGATCAGGACGTAC-3′, IKBKG Sg1 (bottom) 5′-AAACGTACG TCCTGATCTGCTGCC-3′; and IKBKG Sg2 (top) 5′-CACCGCTGCACCATCT CACACAGT-3′, IKBKG Sg2 (bottom) 5′-AAACACTGTGTGAGATGGTGC AGC-3′ were cloned into the vector pSpCas9n (BB)−2A-Puro (PX462) (Addgene). The pSpCas9n (BB)-2A-Puro-IKKγ_gRNAs vector encoded the guide RNA and the Cas9 nuclease with D10A nickase mutant.

U2OS cells (ATCC HTB-96) were seeded in 6-well plates (2 × 105 cells per well) in complete growth medium. The following day, with pSpCas9n (BB)-2A-Puro-RELA/IKKγ_gRNAs and repair template donor plasmids were linearized using BGLII, and cells were transfected using FuGENE HD (Promega) with a transfection reagent to DNA ratio of 3.5 to 1 and a total DNA amount of 4 μg. After 2 weeks, cells were subjected to single-cell sorting into 96-well plates using Beckman Coulter MoFlo Astrios High Speed. Cells underwent clonal isolation and a positive clone was identified via western blot and confirmed by live-cell imaging.

**Western blot analysis**. U2OS cells (parental and expressing EGFP-RelA/IKKγ via CRISPR Knock-in) were cultured for 24 h in complete growth medium. After treatments, cells were lysed in sodium dodecyl sulfate (SDS)-based lysis buffer consisting of 120 mM Tris-Cl, pH 6.8, 4% SDS supplemented with protease, and phosphatase inhibitors at 4 °C for 30 min. Protein extracts were clarified by centrifugation at 4 °C at 12,000 × g for 10 min. Lysate protein levels were quantified by BCA assay (Pierce). Samples were separated by SDS-polyacrylamide gel electrophoresis, 25 μg total protein per lane, then transferred to polyvinylidene difluoride membranes. Blocking was done in 5% milk in TBS for 1 h. Primary antibodies directed at RelA and β-actin (#4764 and #3700, respectively; Cell Signaling Technology), IKKγ, and GAPDH (sc-8330 and sc25778, respectively; Santa Cruz) were diluted (1:1000 for all primary antibodies) in 5% milk in TBS-T and incubated overnight at 4 °C. Conjugated secondary antibodies (LICOR; 1:10,000 dilution) were used in combination with an Odyssey (LI-COR) scanner for detection and quantification of band intensities. Uncropped scans of all western blots are available in the Source Data file.

**Live-cell imaging and analysis**. Live cells were imaged in an environmentally controlled chamber (37 °C, 5% CO₂) on a DeltaVision Elite microscope equipped with a pco.edge sCMOS camera and an Insight solid-state illumination module (GE). U2OS cells expressing FP-RelA/IKKγ were seeded at a density of 25,000 cells/well 24 h prior to live-cell imaging experiments on no. 1.5 glass bottom 96-well imaging plates (Matriplate). For imaging of FP-RelA nuclear translocation, live cells were pretreated with DMSO or the indicated concentrations of compounds for 2 h before exposure to either 100 ng/ml recombinant human TNF (Peprotech) or 100 ng/ml recombinant human IL-1β (Peprotech). Wide-field epifluorescence and differential interference contrast (DIC) images were collected using a ×20 LUCPLFLN objective (0.45 NA; Olympus). Cells were imaged for at least 30 min prior to addition of compounds. For detection of IKKγ puncta, live cells were pretreated with DMSO or the indicated

concentration of compounds for 2 h before exposure to 100 ng/ml TNF. Wide-field epifluorescence and DIC images were collected using a ×60 LUCPLFLN objective. For all treatments, cytokine mixtures were prepared and prewarmed so that addition of 120 µL added to wells results in a final concentration as indicated. Time-lapse images were collected over at least 4 fields per condition with a temporal resolution of 5 min per frame. Quantification of nuclear FP-RelA localization and the formation IKKγ puncta from flat-field and background-corrected images was performed using customized scripts in Matlab and ImageJ.

**Fixed-cell immunofluorescence and analysis.** For fixed-cell measurement of endogenous RelA (Supplementary Fig. 5), U2OS cells were seeded into plastic bottom 96-well imaging plates (Fisher) at 6000 cells/well 24 h prior to treatment. On the day of the experiment, media containing TNF was prepared at 15× the desired concentration for each well. Timing of TNF treatment was planned such that fixation (0, 10, 30, 60, 90, 120 min) occurred simultaneously for all time points at the same time. Prewarmed 15× cytokine mixture was spiked into wells and mixed. Between treatments, the cells remained in environmentally controlled conditions (37 °C and 5% $CO_2$).

At time zero, media was removed from the wells, 185 µL of phosphate-buffered saline (PBS) was used to wash the wells, and wells were incubated at room temperature in 120 µL of 4% paraformaldehyde (PFA) in 1× PBS for 10 min. Wells were then washed 3× for 3 min with 185 µL 1× PBS and then incubated in 120 µL 100% methanol for 10 min at room temperature. Next wells were washed 3× for 3 min in PBS-T (1×PBS 0.1% Tween 20) followed by 120 µL of primary antibody solution (3% bovine serum albumin (BSA) PBS-T, 1:200 dilution of NF-κB p65 F-6 (sc-8008; Santa Cruz)). Plates were wrapped in para-film and left to incubate at 4 °C overnight. The following morning, wells were washed 3× for 5 min in 185 µL PBS-T followed by incubation for 1 h in 120 µL of the secondary antibody solution (3% BSA PBS-T, 1:2000 dilution IgG Alexa Fluor 647 (Cat#A21235, Thermo Fisher)). PBS-T 185 µL was used to wash the wells for 5 min and they were put into 120 µl Hoechst solution (PBS-T, 200 ng/mL Hoechst) for 20 min. Finally, wells were washed for 5 min with PBST and then 185 µL PBS was used to fill the wells and keep the cells hydrated during imaging. Cells were imaged using Delta Vision Elite imaging system at ×20 magnification with a LUCPLFLN objective (0.45 NA; Olympus). Analysis was done using Cell Profiler to segment cells and quantify median nuclear intensity values. Further analysis was performed using custom scripts in MATLAB.

**Permutation tests to assess statistical significance.** For permutation tests, data from the TNF-only and the indicated experimental condition were combined and randomly distributed into Permuted control and Permuted experimental bins without replacement, preserving the size of the original control and experimental data sets. $10^6$ permutations were performed and the difference between the means of Permuted control and Permuted experimental data were calculated for each permutation to generate a histogram. Two-tailed $p$ values were determined by computing the fraction of permuted datasets where $\Delta\text{mean}_{permuted} \geq \Delta\text{mean}_{unpermuted}$ (Supplementary Figs. 7, 8, and 10).

**In vitro IKKβ kinase assay.** We used recombinant activated IKKβ and the IKKtide substrate (Promega, V4502) with the ADP-Glo bioluminescence assay (Promega, V7001) to evaluate the effects of compounds 1, 2, and 3 on IKKβ kinase activity. 1× kinase buffer A (40 mM Tris-HCl pH 7.4, 20 mM $MgCl_2$, 0.1 mg/mL BSA, supplemented with 2 mM $MnCl_2$, 2 mM dithiothreitol, and 100 µM Sodium vanadate) was used to prepare all components of the reaction. All components were prepared in a 96-well plate and transferred to every other well of a 384-well opaque plate (Sigma-Aldrich, CLS3825-10EA) using a multichannel pipet. We prepared a 2.5× ATP/IKKtide substrate mix (62.5 µM ATP mixed with 0.5 µg/µL IKKtide) and a 5× concentration of the indicated concentration of compounds in 0.5% DMSO, maintaining a final DMSO concentration of 0.1% in all reactions. The components of this kinase reaction were added to each well in the following order: 1 µL of 5× compound or buffer only, 2 µL of 100 ng/µL of IKKβ Kinase or buffer, and 2 µL of 2.5× ATP/IKKtide substrate mix. The plate was briefly spun, and the reaction incubated at room temperature for 1 h. Next, 5 µL of ADP-Glo reagent were added to each well, spun, and incubated for 40 min at room temperature. Finally, 10 µL of Kinase Detection Reagent were added to each well and incubated for 30 min at room temperature. Luminescence from each well was measured using an integration time of 500 ms in a M4 microplate reader (SpectraMax). Data from triplicate reactions were extracted and plotted.

**Compound toxicity comparison.** We compared cytotoxicity of the three compounds with Bay 11-7082 (Cayman, 10010266), an inhibitor of the NF-κB pathway at working concentrations of 1–10 µM, using the LIVE/DEAD Cell Imaging Kit (488/570) (Invitrogen, R37601). For each condition, 15,000 U2OS cells were seeded in 200 µL of growth medium in each well of 96-well plate 48 h before microscopy. Next, medium was changed to medium containing DMSO, 10 µM of the indicated compound, or 10 µM of Bay 11-7082 for the indicated duration (2, 16, or 24 h). Before imaging, medium was changed to phenol red-free FluoBrite Dulbecco's modified Eagle's medium (Gibco, A18967–01) containing 300 ng/mL of Hoechst 33342 and 1:10,000 of both Live Green and Dead Red dyes of the LIVE/DEAD Cell Imaging Kit. Cells were

incubated for 60 min and imaged on the Delta Vision Elite imaging system at ×20 magnification with a LUCPLFLN objective (0.45NA; Olympus). Analysis was done using Cell Profiler to segment cells and quantify median nuclear intensity values. Further analysis was performed using custom scripts in MATLAB. Data from biological triplicates were plotted as mean ± SD.

**Reporting summary.** Further information on experimental design is available in the Nature Research Reporting Summary linked to this article.

**Code availability.** Code used to analyze datasets in the current study are available from the corresponding authors on reasonable request.

## Data availability
The data that support the findings of this study are available from the corresponding authors on reasonable request.

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

## Acknowledgements

This work was funded by NIH grants R01-GM097082 to C.J.C. and R35-GM119462 to R. E.C.L. and support to N.A.P. by NSF GRFP-1247842. We acknowledge Zhaofeng Ye for performing the initial molecular docking of compounds against TRAF2. We also thank Anne-Ruxandra Carvunis in addition to members of the Camacho and Lee laboratories for helpful discussion.

## Author contributions

N.A.P., Q.Z., C.J.C., and R.E.C.L. designed the experiments. N.A.P., Q.Z., J.A.C., and D.L. S. conducted the experiments and collected the data. All the authors analyzed the data. N. A.P., J.A.C., C.J.C., and R.E.C.L. contributed to writing the paper. All authors contributed to editing of the paper.

## Additional information

**Competing interests:** The authors declare no competing interests.

