## [Peer Review File · Nature Communications]

Reviewers' comments:

Reviewer #1 (Remarks to the Author):

The manuscript applied a computational methodology that was introduced in another manuscript to identify compounds that modulate NF- κ B signaling pathway for anti-inflammation therapy. The basic idea of the methodology is to preselect hundreds of compounds whose predicted targets are enriched in the NF- κ B signaling pathway, and then use protein-ligand docking for compound prioritization. The concept of network pharmacology (or systems pharmacology) has been explored by many studies. Protein-ligand docking has been widely used in drug discovery. The use of L1000 data to predict drug-target interaction prediction is introduced elsewhere. Thus, the novelty of this manuscript is moderate. The identification of compounds that can modulate NF- κ B signaling pathway is interesting, but their pharmaceutical potential is uncertain. It may be not appealing to broad range of readers. Another concern is that it is not clear how the compounds are objectively selected. It is well-known that protein-ligand docking is not reliable. More evidences are needed to support the reliability and robustness of the presented method.

The manuscript could be improved by addressing two major concerns:

1. The detailed computational results should be presented so that readers have a clear understanding of the scope of proposed methodology. For example, what is the objective threshold to select 501 compounds? What are docking results such as predicted binding free energies for all targets and all 501 compounds?
2. More experiments may be needed to verify the anti-inflammation potency of the selected compounds and show that they do not disrupt the normal function of NF- κ B signaling.

Reviewer #2 (Remarks to the Author):

In this manuscript, Pabon et al use a systems biology approach to predict small molecules that inhibit rate limiting protein-protein interactions in the NF- κ B signaling pathway, and then validate the efficacy of the selected compounds on NF- κ B nuclear localization in live cells. The prediction of inhibitors has been accomplished in the first step by using a random forest classification model to analyze transcriptomic alterations shared by exposure to a library of small molecules (from the LINCS library) and genetic knockdowns of different NF- κ B pathway components. The list of compounds has been then narrowed down, while the mechanisms of action has been inferred, via structural analysis and molecular docking, and experimental validation has been performed using live cell imaging and single-cell analysis.

The combination of analytical methods and experimental validation in live single cells represents a new, interesting and potentially powerful approach to identifying drugs to target pathways that are difficult to inhibit using conventional drug discovery approaches. Such a so-called network-centric concept to drug discovery (rather than target-centric) has been also proposed by others as an alternative strategy to target conventionally undruggable signaling molecules (e.g. RAS signaling) via identifying master regulators of gene expression states. Therefore, the key concept and the overall strategy presented in this manuscript must be of high importance and is very timely. In general, the modeling approach has been well-utilized, and the results are clearly mentioned in most of the places, and the text is logically sound. However, there are a few key points regarding the follow-up validation studies that have not been addressed in the manuscript, so our understanding of mechanisms of action remains incomplete and potential benefit of the identified compounds remains unclear.

Through an interesting integrative approach, the authors find small molecules predicted to target

core protein interactions in the mature TNFR1 complex. They also use thermal shift assays to confirm that the selected compounds interact with TRAF2 in vitro. The authors confirm that such small molecules disrupt the TNF-induced dynamics of NF- κ B signaling. The connection between these findings, however, remains incompletely mapped out. There is no evidence provided in the current study to demonstrate if the impact of these compounds on NF- κ B signaling is actually due to their inhibitory impact on protein-protein interactions in the TNFR1 complex. This is a key point that needs to be addressed:

(a) To confirm the mechanism of action, for example, we may expect that compounds 2 and 3 would exhibit no significant impact on NF- κ B signaling when induced by stimulations that do not engage TRAF2 or TNFR1 complex. Would it be possible to test this hypothesis? Is IL-1 a good option to test that?

(b) I also believe that it is important to rule out the possibility that the impact of these compounds on NF- κ B are due to their potential kinase inhibitory effects. Given that both compounds 2 and 3 are predicted to bind to IKKs and their efficacy is observed only at high concentrations (i.e. 10 μ M), would it be possible that they could act as IKK inhibitors at this high concentration? This might be an important point to address, because compound 1, which is not predicted to bind to IKKs, is not also showing any effect on NF- κ B signaling.

(c) The primary rationale for this study, as presented by the authors, is that current inhibitors of the NF- κ B pathway, e.g. IKK inhibitors, work poorly as therapeutic agents, because the basal activity of these proteins are required for cell survival independent of inflammatory signaling. So, we need new inhibitors, e.g. the ones that target protein-protein interactions in the pathway. One might ask about the performance of the identified compounds (2 and 3) in terms of selective inhibition of inflammatory signaling? Should we expect a lower cytotoxic effect for these compounds relative to IKK inhibitors at concentrations that they inhibit NF- κ B localization to similar levels?

(d) I wonder how informative the thermal shift assay performed in compounds 2 and 3 is for the suggested mechanism of action for these compounds. Would it be possible to make a more rigorous quantitative comparison between the K_d values inferred from the thermal shift assay and compound potencies for inhibition of NF- κ B? Also, are compound potencies for experiments performed in Figures 3 and 4 comparable?

Other comments:

- The statistical analysis tools (random forest classification, etc.), the process and the outcomes are worthy to explain in more detail in the results section of the paper.
- It is difficult to define the statistical significance of single-cell data presented in Figure 3d by using a t test. Given the relatively large number of individual cells in each comparison, a permutation test would be more reasonable.

Dear Reviewers,

We would like to thank you for taking the time to provide insightful and highly constructive comments. Our efforts to address all the concerns raised by the first submission are described below in the point-by-point response. We hope the reviewers will agree that by addressing their comments our revised manuscript has improved significantly, and we hope that it will be considered acceptable for publication in *Nature Communications*. In the revised manuscript, textual changes are marked in blue font. Here we provide a brief overview of alterations to figures and new results provided in additional figures:

- Panel 'd' of Figure 3 now emphasizes the 'AUC' and 'MaxFold', the two most important descriptors of nuclear NFκB dynamics. Other descriptors have been moved to Supplementary figure 6.
- New Fig. 4 shows that compounds 2 and 3 do not inhibit IL1-induced translocation of NF-κB, demonstrating the specificity of compounds 2 & 3 towards the TNF response and the mature TNFR1 complex.
- Supplementary Fig. 1 has been corrected and updated to more accurately reflect the pipeline.
- New Supplementary Fig. 2 presents a scatterplot of docking scores and average Pearson correlations on TNFR1 complex to stratify the 717 compounds predicted to disrupt the NF-κB pathway.
- New supplementary figure 7, related to figure 3 and supplementary figure 5, show permutation histograms used to demonstrate significance of the compounds on the TNF response.
- Permutation histograms are also added to supplementary figure 8, to demonstrate non-significance of compound 1 on the TNF response.
- New supplementary figure 9 shows potency effects for compounds 2 & 3, demonstrating that the IC50 for NFκB dynamics is between 1 and 10uM for both.
- New supplementary figure 10, related to new figure 4, show permutation histograms used to demonstrate non-significance of the compounds' effects on the IL1 response
- New Supplementary figure 11 demonstrates that compounds 1, 2, and 3 are not cytotoxic in contrast with toxicity observed within 16-24 hours for the common NF-κB inhibitor Bay 11-7082.
- New Supplementary figure 12 demonstrates that compounds 1, 2, and 3 do not inhibit *in vitro* kinase activity of IKKβ.
- New Supplementary figure 15 provides additional structural detail for the compounds.

Reviewers' comments:

Reviewer #1 (Remarks to the Author):

The manuscript applied a computational methodology that was introduced in another manuscript to identify compounds that modulate NF-κB signaling pathway for anti-inflammation therapy. The basic idea of the methodology is to preselect hundreds of

compounds whose predicted targets are enriched in the NF- κ B signaling pathway, and then use protein-ligand docking for compound prioritization. The concept of network pharmacology (or systems pharmacology) has been explored by many studies. Protein-ligand docking has been widely used in drug discovery. The use of L1000 data to predict drug-target interaction prediction is introduced elsewhere. Thus, the novelty of this manuscript is moderate. The identification of compounds that can modulate NF- κ B signaling pathway is interesting, but their pharmaceutical potential is uncertain. It may be not appealing to broad range of readers.

We thank the reviewer for their consideration and input. We agree with the reviewer that our approach is integrating a diverse set of ideas to a difficult problem. Systems pharmacology is a broad field but, to our knowledge, no method has been developed to test modulators that target specific signaling pathways as a functional unit. Molecular docking, which has known limitations in ranking docked models, is effectively applied here to predict potential physical interactions among the limited subset of genetic targets.

Most importantly, our recent publication on transcriptomics (Ref ¹) that details a method to identify drug-target interactions is quite different. Namely, the major source of false positives for drug-target interactions in the paper was found to be the fact that the gene profile of a given compound often correlates with that of KO of genes down/up-stream of the target (a consequence of the connectivity in the network). *Here, this novel concept of 'guilt by association' is leveraged for a class of compounds that are identified to be likely disruptors of the pathway.*

The new version of the paper was changed with textual modifications throughout to make the point more clearly. Of particular relevance, in the first results paragraph: *"In the context of a protein interaction network, disruption of a physical target by a drug can cause similar gene expression profiles as inhibition of downstream or upstream genes in the same subnetwork. Hence, a compound that disrupts TRADD or TRAF2 in Fig. 1a might have similar signatures to the knockdown of genes in the pathway such as TNFR1, UBC, or NEMO (see below). Here, we leverage this 'guilt by association', which suggests chemical inhibition acts broadly within a signaling subnetwork (Supplementary Fig. 1), to drug the NF- κ B signaling pathway."*

Regarding the comment, "Another concern is that it is not clear how the compounds are objectively selected. It is well-known that protein-ligand docking is not reliable. More evidences are needed to support the reliability and robustness of the presented method."

We agree with the reviewer that protein-ligand scoring is not always reliable, yet methods are much better at predicting docked poses². Thus, as for any computational method, it can be improved by incorporating additional insight and prior knowledge. With regards to the compounds tested here, Figure 1 shows that compounds were selected based on strong transcriptional correlations with relevant genes in the KEGG and TNFR1 complex for the NF- κ B pathway (see Fig. 1 a and b; also updates to Supplementary Fig. 1). Chemical and transcriptional

similarity between compounds 2 and 3 were viewed as a strong indication that their transcriptional signatures in the NFKB pathway were robust. Finally, we use known co-crystals of TRAF2 protein-protein interactions (PPIs) to visually inspect docked models to eliminate possible false positives that showed problematic poses, or poses that do not disrupt the PPIs. The new Supplementary Fig. 2 demonstrates enrichment of selected compounds in terms of both VINA score³ and correlation with gene expression signatures for compounds and gene knockouts of the TNFR1 complex. Following the advice of the reviewer, transcriptional signatures and docking scores for all 717 compounds predicted to target this pathway are shown.

The manuscript could be improved by addressing two major concerns:

1. The detailed computational results should be presented so that readers have a clear understanding of the scope of proposed methodology. For example, what is the objective threshold to select 501 compounds? What are docking results such as predicted binding free energies for all targets and all 501 compounds?

We thank the reviewer for identifying a confusing typo in the 'Analysis of gene expression data' section of the materials and methods. Compounds are selected based on the genetic prediction that the compounds target a given set of targets (Supplementary Fig. 1). For TRAF2, TRADD and RIPK1, these compounds amount to 717. For details of the computational results please see discussion above, and Supplementary Fig. 2.

2. More experiments may be needed to verify the anti-inflammation potency of the selected compounds and show that they do not disrupt the normal function of NF-κB signaling.

Although the boxplot distributions for descriptors of nuclear RelA dynamics (Fig 3) show the effect of the compounds on single-cell responses, we agree with the reviewer that this is not an ideal way to convey the concept of potency for the selected compounds. Supplementary figure 9 addresses this by plotting the mean and standard deviations of our data set with respect to controls ('DMSO only' and 'TNF without compounds') which relates more directly to more conventional inhibition curves collected from averages in a cell population. The data show that effective doses and the IC50 are between 1-10uM for both compounds 2 & 3.

To demonstrate that the compounds do not disrupt the normal function of NF-κB signaling we performed experiments that stimulate the signaling pathway using interleukin-1 (IL1). IL1 is another inflammatory cytokine that activates NF-κB via the functional IKK complex but is independent of interactions between TRADD and TRAF2. In contrast with our findings for the TNF response, we found that the same concentration of compounds 2 and 3 do not affect IL1-induced activation of the NF-κB system (new Figure 4, and new Supplementary Figure S10). These data demonstrate that the NF-κB system and the functional IKK complex are intact in the presence of the compounds. We furthermore show in new Supplementary Figure S12 that the

compounds do not act on the kinase activity of IKK β , strengthening the argument that direct regulator of NF- κ B signaling is intact. We feel that these significant observations, in addition to other changes described below, demonstrate specificity of our compounds towards the mature TNFR1 complex in the TNF response. Consistent with this experimental observation, we indicate that TRAF6, which is part of the IL1 pathway, does not have any of the residues predicted to interact with our compounds in TRAF2/5.

Reviewer #2 (Remarks to the Author):

In this manuscript, Pabon et al use a systems biology approach to predict small molecules that inhibit rate limiting protein-protein interactions in the NF- κ B signaling pathway, and then validate the efficacy of the selected compounds on NF- κ B nuclear localization in live cells. The prediction of inhibitors has been accomplished in the first step by using a random forest classification model to analyze transcriptomic alterations shared by exposure to a library of small molecules (from the LINCS library) and genetic knockdowns of different NF- κ B pathway components. The list of compounds has been then narrowed down, while the mechanisms of action has been inferred, via structural analysis and molecular docking, and experimental validation has been performed using live cell imaging and single-cell analysis.

The combination of analytical methods and experimental validation in live single cells represents a new, interesting and potentially powerful approach to identifying drugs to target pathways that are difficult to inhibit using conventional drug discovery approaches. Such a so-called network-centric concept to drug discovery (rather than target-centric) has been also proposed by others as an alternative strategy to target conventionally undruggable signaling molecules (e.g. RAS signaling) via identifying master regulators of gene expression states. Therefore, the key concept and the overall strategy presented in this manuscript must be of high importance and is very timely. In general, the modeling approach has been well-utilized, and the results are clearly mentioned in most of the places, and the text is logically sound. However, there are a few key points regarding the follow-up validation studies that have not been addressed in the manuscript, so our understanding of mechanisms of action remains incomplete and potential benefit of the identified compounds remains unclear.

We thank the reviewer for their kind words and encouragement. We have carefully implemented the reviewer's suggested changes and experiments. We hope the reviewer will agree that through textual revisions and the validation studies, our manuscript has improved significantly and more strongly support the mechanism/specificity of the compounds.

Through an interesting integrative approach, the authors find small molecules predicted to target core protein interactions in the mature TNFR1 complex. They also use thermal shift assays to confirm that the selected compounds interact with TRAF2 in vitro. The authors

confirm that such small molecules disrupt the TNF-induced dynamics of NF- κ B signaling. The connection between these findings, however, remains incompletely mapped out. There is no evidence provided in the current study to demonstrate if the impact of these compounds on NF- κ B signaling is actually due to their inhibitory impact on protein-protein interactions in the TNFR1 complex. This is a key point that needs to be addressed:

(a) To confirm the mechanism of action, for example, we may expect that compounds 2 and 3 would exhibit no significant impact on NF- κ B signaling when induced by stimulations that do not engage TRAF2 or TNFR1 complex. Would it be possible to test this hypothesis? Is IL-1 a good option to test that?

We thank the reviewer for this thoughtful suggestion. Ligation of IL1 to its cognate receptor activates the NF- κ B pathway that requires kinase activity from the IKK complex, but independent of TRADD/TRAF2 interactions. It is therefore an ideal suggestion to demonstrate the specificity of our compounds in the IL1 response which induces strong nuclear translocation of the NF- κ B transcription factor. These experiments (Fig. 4 and S10) demonstrate that IL1-induced NF- κ B translocation dynamics are not significantly altered by the presence of compounds 2 and 3. This is an important observation for at least two reasons. First it demonstrates that the NF- κ B signaling system is intact and that bioactivity of our compounds is specific to signaling flux mediated by the mature TNFR1 complex. Secondly, the experiment demonstrates that the compounds do not functionally inhibit kinase activity of the IKK complex in cells when presented at concentrations that limit the TNF-induced NF- κ B response, partially addressing the concern presented in (b). Finally, we note TRAF6, which is part of the IL1 pathway, does not have any of the residues predicted to interact with our compounds in TRAF2/5, providing further support of our proposed mode of action.

(b) I also believe that it is important to rule out the possibility that the impact of these compounds on NF- κ B are due to their potential kinase inhibitory effects. Given that both compounds 2 and 3 are predicted to bind to IKKs and their efficacy is observed only at high concentrations (i.e. 10 μ M), would it be possible that they could act as IKK inhibitors at this high concentration? This might be an important point to address, because compound 1, which is not predicted to bind to IKKs, is not also showing any effect on NF- κ B signaling.

Compounds identified through the computational methodology correlate with many components of the signaling pathway, and although we interpret this as a 'network-level effect' there could still exist secondary interactions that limit IKK kinase activity. Again, we thank the reviewer for this thoughtful suggestion. As we show in Supplementary Figure 12, neither compounds 1, 2, nor 3 inhibit kinase activity of IKK β *in vitro* even at concentrations 10-fold higher than used in cellular assays. This experiment demonstrates that both compounds act through a mechanism different from kinase inhibition.

(c) The primary rationale for this study, as presented by the authors, is that current inhibitors of

the NF- κ B pathway, e.g. IKK inhibitors, work poorly as therapeutic agents, because the basal activity of these proteins are required for cell survival independent of inflammatory signaling. So, we need new inhibitors, e.g. the ones that target protein-protein interactions in the pathway. One might ask about the performance of the identified compounds (2 and 3) in terms of selective inhibition of inflammatory signaling? Should we expect a lower cytotoxic effect for these compounds relative to IKK inhibitors at concentrations that they inhibit NF- κ B localization to similar levels?

We thank the reviewer for suggesting this experiment. We have added data in new Supplementary Figure 11 demonstrating that compounds 1, 2, and 3 are significantly less cytotoxic at both early and late time points in comparison to the common NF- κ B inhibitor, Bay 11-7082. This result increases the relevance of our findings and considered together with results from (a) and (b) strongly suggest that this network-centric approach can identify specific, and therefore less cytotoxic inhibitors of signaling networks.

(d) I wonder how informative the thermal shift assay performed in compounds 2 and 3 is for the suggested mechanism of action for these compounds. Would it be possible to make a more rigorous quantitative comparison between the K_d values inferred from the thermal shift assay and compound potencies for inhibition of NF- κ B? Also, are compound potencies for experiments performed in Figures 3 and 4 comparable?

We acknowledge that the thermal shift is a weak surrogate for K_d and that these values are derived from experiments performed at 55-60C, whereas affinities are normally derived at room temperature or 4C. Also, because these proteins are large oligomeric structures, thermal shift is difficult to translate into a relevant biophysical parameter such as K_d . Instead, we provide a direct evaluation of potency using the live-cell data. Previously, we've emphasized the dynamics of single-cell responses (Fig. 3). New Supplementary figure 9 shows the equivalent assessment of potency as would be observed from averaging the effects of inhibition across all cells in a population (which is the more standard presentation for biochemical data that does not consider dynamics and single-cell variability).

Regarding the potency for IKK inhibition. Unlike the NF- κ B reporter where we have exhaustively defined the descriptors of dynamics that carry information in the signaling system (Lee et al., Molecular Cell, 2014; Zhang, Gupta, et al. Cell Systems, 2017), from a quantitative standpoint the IKK reporter is comparatively uncharacterized. These data are therefore intended to showcase qualitative differences that can be clearly observed i.e. the presence or absence of IKK puncta which represent the penultimate step in forming the mature TNFR1 complex. We are concerned that making claims to potency would require characterization of the IKK reporter system that are significantly beyond the scope of the manuscript i.e. at moderate concentrations some other property of the spots beyond their presence/absence may be important, such as the life span or size of individual spots among other features, and it's non-trivial to define the meaning of 'potency' for this system. Although studies to rigorously define

relevant descriptors of IKK dynamics are ongoing projects in the lab, they are years from completion. Instead, for the IKK experiments we simply used the same 10uM concentration of Compounds 2 and 3 as shown in the figure panel to demonstrate a qualitative, yet stark demonstration that the TNFR1 complex fails to recruit IKK and become mature when cells are stimulated with TNF.

Other comments:

- The statistical analysis tools (random forest classification, etc.), the process and the outcomes are worthy to explain in more detail in the results section of the paper.

We've made a significant number of edits to the manuscript that increase clarity. These highlight the training of the model in addition to the overall flow of compound selection via Pearson correlation and docking scores (see New Supplementary Fig. 2). Since our first submission the method for single drug-target interactions was accepted for publication (REF ¹). This paper contains a detailed explanation of the random forest classification technique. In addition to the methods and explanation provided in the revised manuscript, we have included the citation where possible to direct readers to the appropriate resource with even more description.

- It is difficult to define the statistical significance of single-cell data presented in Figure 3d by using a t test. Given the relatively large number of individual cells in each comparison, a permutation test would be more reasonable.

We thank the reviewer for the suggested analysis. In re-visiting the primary data for the descriptors of NF- κ B dynamics we discovered a minor plotting error that did not affect the results and has now been corrected. In the revised manuscript we now use permutation tests to show statistical significance for all live-cell studies of NF- κ B dynamics. These have subtly altered the results but still clearly emphasize the impact of compounds 2 and 3 on NF- κ B dynamics. These changes are reflected in alterations to Figure 3 and Supplementary figure 6. Distributions resulting from the permutation tests for all statistical comparisons between descriptors are shown in supplementary figures 7, 8, and 10 for, and the implementation is described in the methods.

- 1 Pabon, N. A. *et al.* Predicting protein targets for drug-like compounds using transcriptomics. *PLoS Comput Biol* **14**, doi:10.1371/journal.pcbi.1006651 (2018).
- 2 Ye, Z., Baumgartner, M. P., Wingert, B. M. & Camacho, C. J. Optimal strategies for virtual screening of induced-fit and flexible target in the 2015 D3R Grand Challenge. *J Comput Aided Mol Des* **30**, 695-706, doi:10.1007/s10822-016-9941-0 (2016).
- 3 Trott, O. & Olson, A. J. AutoDock Vina: improving the speed and accuracy of docking with a new scoring function, efficient optimization, and multithreading. *J Comput Chem* **31**, 455-461, doi:10.1002/jcc.21334 (2010).

REVIEWERS' COMMENTS:

Reviewer #1 (Remarks to the Author):

Thank authors for addressing the issues raised and improving the manuscript.

Reviewer #2 (Remarks to the Author):

The points raised in the previous round of review have been satisfactorily addressed. The authors have performed additional experiments (using IL-10) to address previous concerns regarding potential modes of action for the identified compounds. The statistical analysis and description of the methods has been improved. Overall, the revised manuscript represents an interesting and network-level approach to drug discovery, which integrates a diversity of powerful tools from transcriptomic and structural modeling to live-cell imaging and single-cell analysis.

Mohammad Fallahi-Sichani

We thank both reviewers for their assistance reviewing our manuscript and for providing helpful suggestions.

RECL and CJC

REVIEWERS' COMMENTS:

Reviewer #1 (Remarks to the Author):

Thank authors for addressing the issues raised and improving the manuscript.

Reviewer #2 (Remarks to the Author):

The points raised in the previous round of review have been satisfactorily addressed. The authors have performed additional experiments (using IL-10) to address previous concerns regarding potential modes of action for the identified compounds. The statistical analysis and description of the methods has been improved. Overall, the revised manuscript represents an interesting and network-level approach to drug discovery, which integrates a diversity of powerful tools from transcriptomic and structural modeling to live-cell imaging and single-cell analysis.

Mohammad Fallahi-Sichani